# Fast Estimation of the Spectral Optical Properties of Rabbit Pancreas and Pigment Content Analysis

Inês Soraia Martins [1,2], Hugo Filipe Silva [1,2], Valery Victorovich Tuchin [3,4,5] and Luís Manuel Oliveira [2,6,*]

1 Faculty of Engineering, University of Porto, FEUP, Rua Dr. Roberto Frias, 4200-465 Porto, Portugal; isdma@isep.ipp.pt (I.S.M.); hfdis@isep.ipp.pt (H.F.S.)
2 Center of Innovation in Engineering and Industrial Technology, ISEP, Rua Dr. António Bernardino de Almeida 431, 4249-015 Porto, Portugal
3 Science Medical Center, Saratov State University, 83 Astrakhanskaya Str., 410012 Saratov, Russia; tuchinvv@mail.ru
4 Laboratory of Laser Molecular Imaging and Machine Learning, Tomsk State University, 36 Lenin's Av., 634050 Tomsk, Russia
5 Laboratory of Laser Diagnostics of Technical and Living Systems, Institute of Precision Mechanics and Control, FRC "Saratov Scientific Centre of the Russian Academy of Sciences", 24 Rabochaya, 410028 Saratov, Russia
6 Physics Department, School of Engineering, Polytechnic Institute of Porto, Rua Dr. António Bernardino de Almeida 431, 4249-015 Porto, Portugal
* Correspondence: lmo@isep.ipp.pt

**Abstract:** The pancreas is a highly important organ, since it produces insulin and prevents the occurrence of diabetes. Although rare, pancreatic cancer is highly lethal, with a small life expectancy after being diagnosed. The pancreas is one of the organs less studied in the field of biophotonics. With the objective of acquiring information that can be used in the development of future applications to diagnose and treat pancreas diseases, the spectral optical properties of the rabbit pancreas were evaluated in a broad-spectral range, between 200 and 1000 nm. The method used to obtain such optical properties is simple, based almost on direct calculations from spectral measurements. The optical properties obtained show similar wavelength dependencies to the ones obtained for other tissues, but a further analysis on the spectral absorption coefficient showed that the pancreas tissues contain pigments, namely melanin, and lipofuscin. Using a simple calculation, it was possible to retrieve similar contents of these pigments from the absorption spectrum of the pancreas, which indicates that they accumulate in the same proportion as a result of the aging process. Such pigment accumulation was camouflaging the real contents of DNA, hemoglobin, and water, which were precisely evaluated after subtracting the pigment absorption.

**Keywords:** pancreas tissue; optical properties; scattering coefficient; absorption coefficient; pigment detection

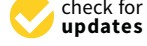



## 1. Introduction

Since the middle of the XXth century the research and application of optical methods in clinical practice have evolved greatly. Light is a fast tool that can aid in the establishment of a precise diagnosis, does not cause side effects, can be focused with great precision to be used as a scalpel in the removal of damaged or diseased tissues or induce photochemical treatment effects in a noninvasive or minimally invasive way [1]. Due to the existence of various diagnostic or treatment applications in clinical practice that work in different spectral ranges from the ultraviolet (UV) to the terahertz (THz) [2], a particular field of interest in biophotonics is the determination of the spectral optical properties of tissues in a wide wavelength ($\lambda$) range. Such estimation of the optical properties of biological tissues in a wide spectral range is highly important for the improvement of actual technologies

and for the development of new optical approaches to be used in diagnostic and treatment procedures [2,3].

The optical properties of biological tissues can be considered as their identity card since they are particular for every tissue and differences in the internal composition, physiology, or morphology of tissue will induce changes in its optical properties. The characterization of biological tissues is in general made through a set of three fundamental optical properties that describe the absorption, the scattering, and the scattering directionality of photons inside [4]. Such a set of optical properties is composed by the absorption coefficient ($\mu_a$), the scattering coefficient ($\mu_s$), and the scattering anisotropy factor ($g$), which are considered the most fundamental [5], but other properties such as the reduced scattering coefficient ($\mu'_s$) or the light penetration depth ($\delta$) can be considered [6–13].

Using current technologies, it is possible to measure broadband spectra from ex vivo tissue samples, such as the total transmittance ($T_t$), the total reflectance ($R_t$), and the collimated transmittance ($T_c$) spectra, which can be used in estimation or calculation algorithms to obtain the spectral optical properties of tissue for a wide $\lambda$-range [14]. Some previous publications report the spectral optical properties for various tissues, such as peritoneal tissues [11], human colon tissues [12,15], human skin and subcutaneous tissues [13], adipose tissues [16], cranial bone [17], stomach mucosa [8], human sclera [10], normal and pathological colorectal mucosa [6,15,18], colorectal muscle [19] and human normal and pathological liver [20,21]. Although the optical properties of these tissues have already been studied in a wide spectral range, there are many other tissues where such study has not yet been made.

The pancreas is an organ with high importance in the human and animal bodies, but one of the less studied with optical methods. The pancreas is responsible for the production of insulin in the body, which prevents the occurrence of diabetes [22]. The incidence of pancreatic cancer in humans, although rare, has been growing over past years [23]. Since there is a significant lack of diagnostic and treatment methods, this type of cancer is highly lethal, and although several researchers have tried to develop new therapeutic approaches for pancreatic cancer, the results have been disappointing [24]. Pancreatic cancer that presents the most lethal malignant neoplasia in humans is the ductal adenocarcinoma, a pathology that is in general detected at an advanced stage of development. Such a fact reduces the life expectancy to one year after the diagnosis is established [25]. Although this is the deadliest of the pancreatic cancers, there are other types, such as the acinar cell carcinoma, the pancreatoblastoma, the solid pseudopapillary neoplasia, the pancreas neuroendocrine tumor, the insulinoma, the vipoma, the glucagonoma, the somatostatinoma, or the pancreatic neuroendocrine tumor [26]. With such a high number of pancreatic cancer varieties, it becomes urgent to identify risk factors as well as to develop procedures to obtain an early-stage diagnosis. This makes it possible to implement preventive measures on time.

The application of optical technologies to detect cancer or other pathologies of the pancreas can be highly helpful, but the interest in studying the pancreas with optical technologies is only now emerging. A recent paper has reported the differences between the blood microcirculation in normal and in diseased rat pancreas with induced alloxan diabetes [27]. Another recent paper presents discrete experimental refractive index (RI) data and the calculated dispersion between 400 and 1000 nm for the rabbit pancreas [28]. Although these two studies are most valuable for the application of optical methods to the pancreas, the evaluation of its optical properties over a wide spectral range is still necessary to plan further diagnostic optical methods.

With the objective of studying the spectral optical properties of the pancreas and obtain detailed information about its internal composition, tissue samples from the rabbit pancreas were used to conduct the present study. The methodology used to prepare the tissues, to conduct the experimental measurements and to calculate the spectral optical properties of the pancreas is described in Section 2. The experimental and calculated results are presented in Section 3 along with the discussion of the information acquired.

## 2. Materials and Methods

The present study consisted of the calculation of the spectral optical properties of the pancreas. To conduct such a study, it was necessary to measure some types of spectra from excised tissue samples of the rabbit pancreas, which were used in the necessary calculations. This research is accordingly with the Declaration of Helsinki and was approved by the research review board in biomedical engineering of the Center of Innovation in Engineering and Industrial Technology (CIETI), in Porto, Portugal. Such approval has the number CIETI/Biomed_Research_2021_01. According to the methodology adopted in this study, the following subsections describe the sample collection and preparation to perform the necessary measurements and the calculations made to obtain such spectral optical properties.

### 2.1. Tissue Collection and Preparation

Five adult grey rabbits, of similar age near 36 months, were acquired from a breeder near Porto (Portugal) that sells them for consumption. Since the rabbit pancreas is a small organ, it was only possible to collect two samples from the pancreas of each animal to use in the spectral measurements. On the day before conducting a set of experimental measurements, one rabbit was sacrificed and the pancreas was frozen for 12 h, at $-20\,^{\circ}$C. After this period, two tissue samples were sliced from the pancreas using a cryostat (Leica$^{TM}$, Wetzler, Germany, model CM1860 UV). Ten samples in total, two from each rabbit, were prepared to conduct the spectral measurements on different days. These samples had an approximated circular form ($\varphi \sim 1$ cm) and a fixed thickness of 0.5 mm. The samples were all submitted to three types of spectral measurements, which are described in Section 2.2. After slicing the samples with the cryostat, they were immersed in saline for ten minutes to regain the in vivo hydration and warm up to reach the room temperature of $20\,^{\circ}$C, the temperature at which the measurements were made.

### 2.2. Spectral Measurements

In calculating the spectral optical properties of the rabbit pancreas, three types of spectral measurements were necessary. These spectral measurements were made using the setups presented in Figure 1.

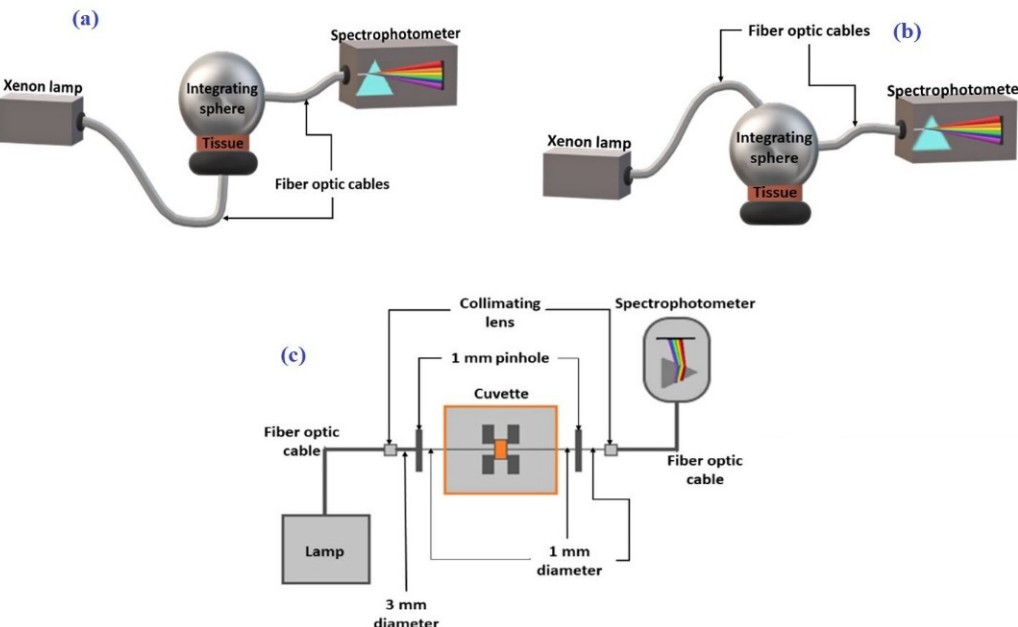

**Figure 1.** Experimental setups to measure the $T_t$ (**a**), $R_t$ (**b**) and $T_c$ (**c**) spectra.

Considering the setup to measure the $T_t$ spectra that is presented in Figure 1a, a broad-band and high-power pulsed Xenon lamp was used as the light source. The Xenon lamp is connected to the spectrophotometer by a Y-cable, which provides power to the lamp and allows to control the flash emission rate through the software up to a limit of 150 Hz. The beam from this lamp is introduced into an optical fiber cable that delivers it below the tissue sample. A collimating lens at the sample side of the optical fiber guarantees a 6 mm collimated beam at the lower side of the tissue sample. The sample is placed just below the sample port of the integrating sphere and over a black plastic base with a 6 mm hole in the center (see Figure 1a) to allow the light beam to reach the sample. The beam crosses the sample before entering the integrating sphere, where it is submitted to several reflections and integration, before being delivered to the spectrometer through another optical fiber cable.

Using a similar setup to the one used to measure the $T_t$ spectra, the $R_t$ spectra were measured using the setup presented in Figure 1b. The difference between the two setups is that, in the case of $R_t$ measurements, the incident beam is reflected at the top surface of the tissue sample before being reflected and integrated inside the sphere. In this reflection mode, such incident beam enters the integrating sphere at 8° with the vertical axis of the sphere. In these measurements, the tissue sample is also placed at the sample port of the integrating sphere and over a black plastic base as in the $T_t$ measurements. The difference is that to measure the $R_t$ spectra, the 6 mm hole in the black plastic below the sample is closed to avoid contamination of the reflectance measurements with transmitted light (see Figure 1b).

Considering the $T_c$ setup that is presented in Figure 1c, a beam from a broad-band deuterium-halogen lamp is delivered to the left side of the sample cuvette through an optical fiber cable, a collimating lens and a pinhole, which collimate and reduce the beam diameter to 1 mm. The unscattered transmitted beam is collected on the right side of the sample cuvette by another pinhole and lens to introduce it into another optical fiber cable to be delivered to the spectrophotometer. Ten spectra were collected between 200 and 1000 nm from the tissue samples with each setup represented in Figure 1. Those spectra were submitted to the calculations described in Section 2.3 so that the spectral optical properties of the pancreas could be obtained.

*2.3. Calculations*

Once all experimental spectra were acquired, calculations were initiated to obtain the spectral optical properties of the pancreas between 200 and 1000 nm. The calculation procedure had the following sequence:

(a)  $\mu_a(\lambda)$ was calculated from the sample thickness ($d$ = 0.5 mm) and the $T_t$ and $R_t$ spectra, using Equation (1) [6,29]:

$$\mu_a(\lambda) = \frac{\left[1 - \left(\frac{T_t(\lambda) + R_t(\lambda)}{100}\right)\right]}{d}. \tag{1}$$

Such a calculation was made 10 times to obtain the mean and standard deviation (SD) of $\mu_a(\lambda)$;

(b)  The pancreas dispersion ($n_{\text{tissue}}(\lambda)$) was calculated from $\mu_a(\lambda)$, using the Kramers–Kronig (K–K) relations—Equations (2) and (3) [30,31]. First, Equation (2) was used to calculate the imaginary part of the tissue dispersion ($\kappa(\lambda)$) from $\mu_a(\lambda)$ [29–31]:

$$\kappa(\lambda) = \frac{\lambda}{4\pi}\mu_a(\lambda). \tag{2}$$

The second step of this calculation was made to obtain $n_{\text{tissue}}(\lambda)$ from $\kappa(\lambda)$ through Equation (3) [29–31]:

$$n_{\text{tissue}}(\lambda) = 1 + \frac{2}{\pi}\int_0^\infty \frac{\lambda_1}{\Lambda} \times \frac{\lambda_1}{\Lambda^2 - \lambda_1^2}\kappa(\Lambda)d\Lambda, \tag{3}$$

where $\Lambda$ represents the integrating variable over the spectral range of interest and $\lambda_1$ is a fixed $\lambda$ that can be adjusted for a better vertical matching of the calculated dispersion to a Cornu-type dispersion, which was calculated based on discrete experimental RI data that was available from a previous study [28]. Such a calculation was also made 10 times to obtain the mean and SD of $n_{\text{tissue}}(\lambda)$;

(c) To calculate $\mu_s(\lambda)$ for the same spectral range, $d$, $\mu_a(\lambda)$ and the $T_c$ spectrum ($T_c(\lambda)$) were used in the Bouguer–Beer–Lambert law [3,4,6]:

$$\mu_s(\lambda) = -\frac{ln\left[\frac{T_c(\lambda)}{100}\right]}{d} - \mu_a(\lambda), \tag{4}$$

where the $T_c(\lambda)$ is normalized by a factor of 100 to be represented on a 0 to 1 scale. This calculation was also made 10 times and each of the calculated spectra were fitted with a curve described by Equation (5), which is indicated in literature [32] as suitable to describe the $\lambda$-dependencies for $\mu_s$ and for $\mu'_s$ in the deep-UV to the near-infrared range. Previous studies with other tissues have demonstrated that $\mu_s$ and $\mu'_s$ have good agreement in the deep-UV, but with the increase in the wavelength, they show increasing differences up to the near-infrared [3,21,29].

The 10 resulting curves were used to calculate the mean and SD of $\mu_s(\lambda)$

$$\mu_s(\lambda) \text{ or } \mu'_s(\lambda) = a'\left(f_{\text{Ray}}\left(\frac{\lambda}{500 \text{ (nm)}}\right)^{-4} + (1 - f_{\text{Ray}})\left(\left(\frac{\lambda}{500 \text{ (nm)}}\right)^{-b_{\text{Mie}}}\right)\right), \tag{5}$$

where $a'$ represents the $\mu_s$ or the $\mu'_s$ of the pancreas at 500 nm, $f_{\text{Ray}}$ is the Rayleigh scattering fraction and $b_{\text{Mie}}$ is the mean size of the particles responsible for the Mie scattering [3];

(d) Regarding $\mu'_s$ there is no mathematical relation to calculate it directly from spectral measurements, but its $\lambda$-dependence is well described by a combination of the Rayleigh and Mie scattering regimes, as described by Equation (5) [32]. This way, 10 sets of inverse adding-doubling (AD) [33] simulations were made at every 50 nm between 200 and 1000 nm. Such simulations were performed with the code available at the website of the Oregon Medical Laser Centre [34]. As a result of these simulations, 10 curves described by Equation (5) were calculated to fit the various sets of generated $\mu'_s$ values. The mean and SD of these curves were calculated;

(e) Using the 10 calculated spectra of $\mu_s$ and $\mu'_s$, Equation (6) was used to calculate $g(\lambda)$, which is valid in the diffusion approximation [3,4,6–13]:

$$\mu'_s = \mu_s \times (1 - g). \tag{6}$$

Mean and SD for $g(\lambda)$ were obtained from the 10 calculated spectra;

(f) Using the 10 calculated spectra of $\mu_a$ and $\mu'_s$, Equation (7) was used to calculate $\delta(\lambda)$, which is also valid in the diffusion approximation [3,4,6–13]:

$$\delta = \frac{1}{\sqrt{3\mu_a\left(\mu_a + \mu'_s\right)}}. \tag{7}$$

The 10 spectra of $\delta$ that resulted from these calculations were used to obtain the mean and SD spectrum of the light penetration depth.

At the end of these calculations, and performing a careful analysis of the mean $\mu_a(\lambda)$ it was verified that the pancreas, besides proteins, DNA, and blood, contains pigments. A set of simple calculations to identify those pigments and retrieve their content was made. These final calculations and all the results of the present study are presented in Section 3.

## 3. Results

According to the methodology described in Section 2, the present study started with the measurements of the spectra from the tissue samples. Since ten spectra were acquired from the ten tissue samples with each of the experimental setups represented in Figure 1, Figure 2 presents the mean and SD for the $T_t$, $R_t$, and $T_c$ spectra.

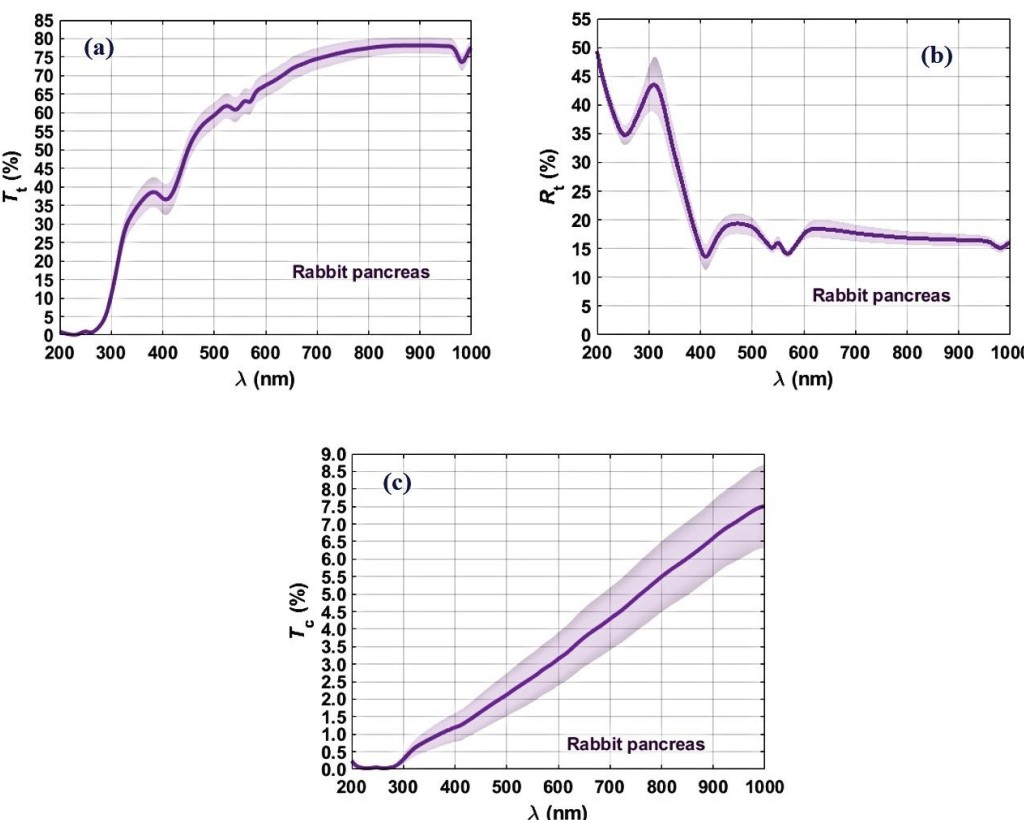

**Figure 2.** Mean and SD of the $T_t$ (**a**), $R_t$ (**b**) and $T_c$ (**c**) spectra.

The mean spectra presented in Figure 2 show typical $\lambda$-dependencies: $T_t$ and $T_c$ increase with $\lambda$ and $R_t$ shows a decreasing tendency from the UV to the NIR. Regarding the SD, we see that in general it is not high, showing good agreement between the spectra measured from the ten samples in each case. Considering the $T_t$ and $T_c$ spectra in Figure 2, the SD is practically null for $\lambda$ shorter than 300 nm due to the strong absorption bands of proteins, DNA/RNA and hemoglobin that occur, respectively, at 200–230, 260 and 274 nm [35–37]. The spectra in Figure 2 show higher SD values in the visible-NIR range due to different transmittance or reflectance between the various samples used in the measurements. The $T_t$ and $R_t$ spectra also show the absorption bands of hemoglobin at 415 nm (Soret band) and 540/570 nm (Q bands), and the band of water at 980 nm [37].

Following the procedure described in Section 2.3, using ten pairs of $T_t$ and $R_t$ spectra in Equation (1), ten $\mu_a$ spectra were calculated. Figure 3 presents the mean and SD that resulted from these calculations.

Figure 3 shows absorption bands at various $\lambda_s$. At 260 nm we see the DNA/RNA band; the Soret band of hemoglobin is seen at 415 nm, its Q bands are visible at 540 and 570 nm, and the absorption band of water is observed at 980 nm. In addition to the identification of the main absorbers in the pancreas, the mean spectrum presented in Figure 3 shows a decreasing baseline. Such behavior indicates the presence of some broad-band absorbers in the pancreas. Further analysis to identify such absorbers will be performed below, after presenting the other spectral optical properties of the pancreas.

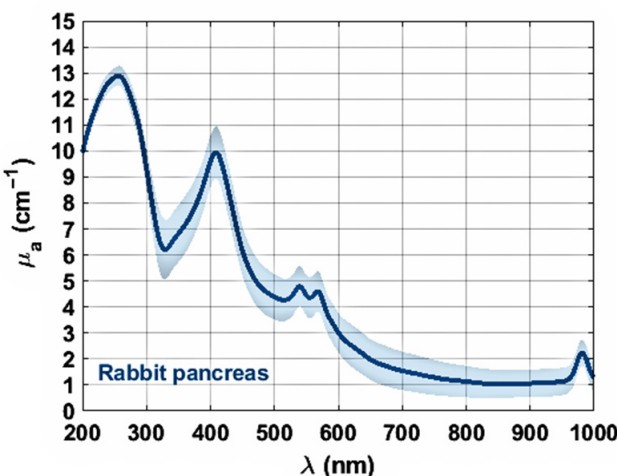

**Figure 3.** Mean and SD of the $\mu_a(\lambda)$ of the rabbit pancreas.

After obtaining $\mu_a(\lambda)$, it was used in the K–K relations (Equations (2) and (3)) to obtain $n_{\text{tissue}}(\lambda)$. Such a calculation was optimized by adjusting $\lambda_1$ in Equation (3) to match $n_{\text{tissue}}(\lambda)$ to the Cornu-type dispersion that was previously calculated from discrete RI measurements [28]. These calculations were made 10 times and the mean results are presented in Figure 4.

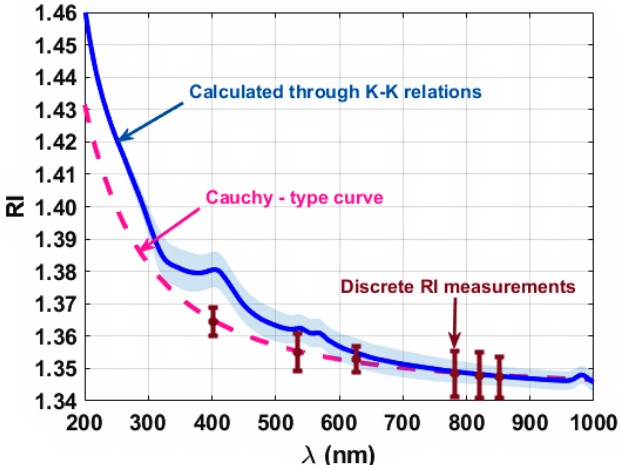

**Figure 4.** Mean and SD of the $n_{\text{tissue}}(\lambda)$ as calculated through K–K relations (blue line), discrete RI measurements (brown) and calculated Cornu dispersion for the rabbit pancreas (pink-dashed curve).

To calculate $\mu_s(\lambda)$, the individual $\mu_a$ spectra were used in Equation (4), along with the experimental $T_c$ spectra and the sample thickness ($d$ = 0.5 mm). Such a calculation was also made ten times, leading to the mean and SD for $\mu_s(\lambda)$ that is represented in Figure 5. To obtain the $\lambda$-dependence for $\mu'_s$ and according to the procedure described in Section 2.3, inverse AD simulations were performed for $\lambda_s$ at each 50 nm between 200 and 1000 nm. Such simulations were performed ten times for each $\lambda$, leading to ten sets of $\mu'_s$ values, which were fitted with curves described by Equation (5). The resulting mean and SD for $\mu'_s(\lambda)$ are also represented in Figure 5.

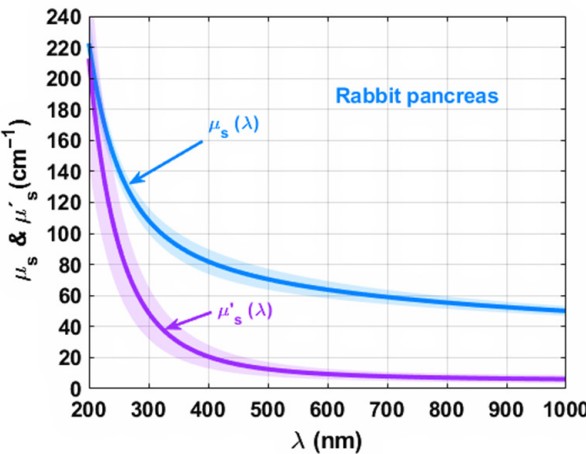

**Figure 5.** Mean and SD of $\mu_s(\lambda)$ and $\mu'_s(\lambda)$ for the pancreas.

The mean curves presented in Figure 5 for $\mu_s(\lambda)$ and $\mu'_s(\lambda)$ are described by Equations (8) and (9), respectively.

$$\mu_s(\lambda) = 70.6\left(0.04431\left(\frac{\lambda}{500\ (\text{nm})}\right)^{-4} + (1-0.04431)\left(\left(\frac{\lambda}{500\ (\text{nm})}\right)^{-0.4358}\right)\right), \quad (8)$$

$$\mu'_s(\lambda) = 12.6\left(0.4113\left(\frac{\lambda}{500\ (\text{nm})}\right)^{-4}\right.$$
$$\left. +(1-0.4113)\left(\left(\frac{\lambda}{500\ (\text{nm})}\right)^{-0.3618}\right)\right). \quad (9)$$

The fitting of the mean data in the curves presented in Figure 5 was obtained with an R-square value equal to 1 in both cases. Both curves that are presented in Figure 5 and that are described by Equations (8) and (9) show a combination of the Rayleigh and Mie scattering regimes in the pancreas [32]. From Figure 5, it is seen that these curves tend to match each other at low values of $\lambda$ and show increasing differences as $\lambda$ increases. Such behavior was already seen in other studies with human colorectal mucosa tissues [6] and with human liver tissues [21]. Figure 5 also shows that the SD of $\mu'_s(\lambda)$ presents a higher magnitude than one of the $\mu_s(\lambda)$, but such difference is related to greater variability of results obtained in the inverse AD simulations for each $\lambda$.

Considering the individual spectra that were calculated for $\mu_s(\lambda)$ and $\mu'_s(\lambda)$, Equation (6) was used to obtain the ten spectra of the $g$-factor. The mean and SD of these calculations are represented in Figure 6.

The mean curve for $g$ in Figure 6 shows the typical $\lambda$-dependence for biological tissues—it grows with the $\lambda$ from very low values at the deep-UV to values close to unity in the visible-NIR region [32]. Such $\lambda$-dependence has been previously obtained with this calculation method for other tissues [6,21]. The magnitude of the SD represented in Figure 6 is maintained as almost constant in the entire spectral range, with the exception of the range between 200 and 250 nm, where major variability in the generated $\mu'_s$ values contributes to higher differences between the calculated $g$ curves. The mean spectrum for the $g$-factor that is represented in Figure 6 is described by Equation (10):

$$g(\lambda) = 0.8549 \times e^{(2.724 \times 10^{-5} \cdot \lambda)} - 5.891 \times e^{(-9.769 \times 10^{-3} \cdot \lambda)}. \quad (10)$$

The fitting of the mean $g(\lambda)$ in Figure 6 with Equation (10) was obtained with an R-square value of 0.9999.

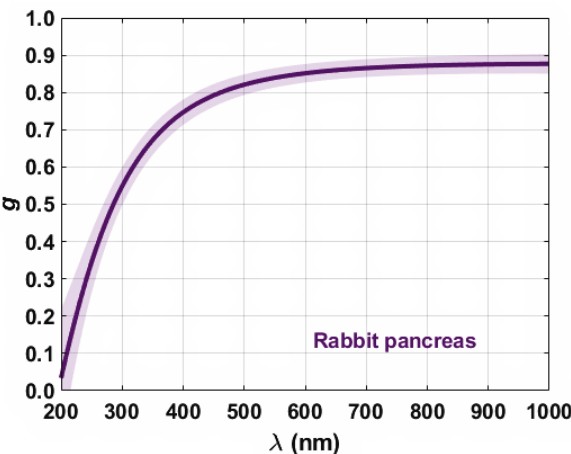

**Figure 6.** Mean and SD of $g(\lambda)$ for the pancreas.

The last optical property that was calculated was $\delta$. Following the calculation procedure described in Section 2.3, using ten pairs of $\mu_a / \mu'_s$ spectra in Equation (7), ten spectra were calculated for $\delta$. Figure 7 presents the mean and SD of these calculations.

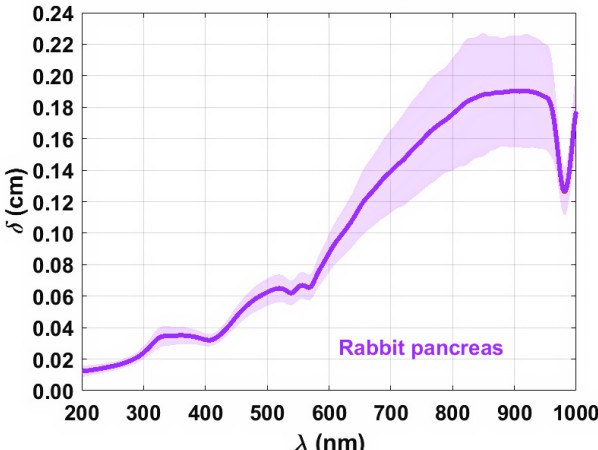

**Figure 7.** Mean and SD of $\delta\,(\lambda)$ for the pancreas.

Once again, the $\lambda$-dependence presented in Figure 7 is according to the description presented in the literature [32] for biological tissues, since $\delta$ increases with $\lambda$ from the deep-UV to the NIR. Such a spectrum also shows the absorption bands of hemoglobin and water due to the contribution of $\mu_a$ in the calculation of $\delta$. The spectrum presented in Figure 7 shows that clinical applications that work in the NIR are preferable to the ones that work in the UV, where minimal penetration depth is observed. These low values observed for $\delta$ in the deep-UV range are due to the combination of strong absorption bands from tissue absorbers such as proteins and DNA with the high scattering that tissues present in this spectral range [38]. To overcome such high attenuation in the UV range, biocompatible optical immersion clearing treatments can be applied, as reported in various publications [38–43]. Recent studies have demonstrated that light scattering can be strongly reduced in the UV range with the application of such treatments [40–42]. In fact, some studies have demonstrated that the magnitude of the transparency created in the visible-NIR range depends on the osmolarity of the optical clearing agent (OCA) used [43,44], and others showed that the transparency efficiency in the deep-UV is considerably higher than in the visible-NIR range [40–42,45].

After calculating all the spectral optical properties of the pancreas, we returned to the calculated $\mu_a$ spectrum to perform further analysis. When discussing the mean spectrum that is presented in Figure 3, we referred to the fact that the decreasing baseline that it

presents is evidence that some broad-band absorbers are contained in the pancreas. In agreement with what has been achieved in previous studies with other tissues [6,29], we considered that those absorbers are pigments, namely melanin, and lipofuscin. Literature indicates that both melanin and lipofuscin accumulate in tissues during the aging process [46–48]. Searching for the $\mu_a$ spectra of melanin and lipofuscin, we found that melanin has an exponentially decreasing absorption with increasing $\lambda$ between 200 and 1000 nm [49], and that lipofuscin has its absorption concentrated between 200 and 600 nm [50]. Collecting numerical values from the graphical representations of the $\mu_a$ spectra of melanin and lipofuscin in Refs. [49,50], we reconstructed their absorption spectra, as represented in Figure 8.

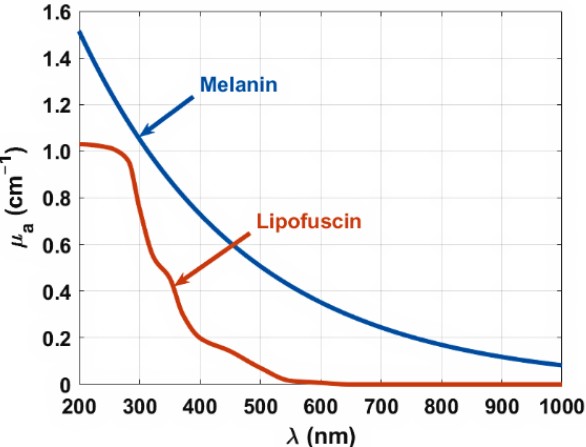

**Figure 8.** Spectral curves of $\mu_a(\lambda)$ for melanin and lipofuscin.

Several trials were made to reconstruct the baseline seen in the $\mu_a(\lambda)$ of the pancreas in Figure 3 by combining both spectra presented in Figure 8. The perfect combination was found to be described as:

$$\mu_{a-\text{pigment}}(\lambda) = 3.76 \times M(\lambda) + 3.65 \times L(\lambda), \tag{11}$$

where $M(\lambda)$ represents the $\mu_a(\lambda)$ of melanin and $L(\lambda)$ represents the $\mu_a(\lambda)$ of lipofuscin as represented in Figure 8. Equation (11) shows that the contents of melanin and lipofuscin are similar in the pancreas, which indicates that both pigments accumulate in the pancreas in the same proportion due to tissue aging.

After obtaining the combination presented in Equation (11) to reconstruct the baseline in the $\mu_a(\lambda)$ of the pancreas, $\mu_{a\text{-pigment}}(\lambda)$ was subtracted from $\mu_a(\lambda)$ to obtain a horizontal baseline. Figure 9 presents the $\mu_a(\lambda)$ of the pancreas before and after subtracting the pigment contribution and also $\mu_{a\text{-pigment}}(\lambda)$, as described by Equation (11). Figure 9 also presents the absorption ratios at the center of the main bands: 260 nm for DNA/RNA, 415 nm for the hemoglobin Soret band, 555 nm for the middle of the Q-bands, and 980 nm for water (see the black double-arrow lines in Figure 9). The values for these ratios that are presented in blue were calculated considering the $\mu_{a\text{-pigment}}(\lambda)$ as the baseline, while the ones presented in green were calculated considering the black horizontal line as the baseline. Such horizontal baseline is in agreement with the minimum value of the corrected spectra, which occurs simultaneously at 200 and 820 nm.

According to the various absorption ratios presented in Figure 9, we see that the ones that were calculated before the subtraction of the $\mu_{a\text{-pigment}}(\lambda)$ from the mean $\mu_a(\lambda)$ of the pancreas are deceiving. Those values do not represent the true DNA/RNA, hemoglobin and water content in the pancreas. When calculating the same ratios after performing the subtraction of the $\mu_{a\text{-pigment}}(\lambda)$, the corrected $\mu_a(\lambda)$ of the pancreas shows increased ratios both for DNA/RNA and hemoglobin. In the case of water, the absorption ratio at 980 nm decreases to about $\frac{1}{2}$ of the value before the subtraction operation.

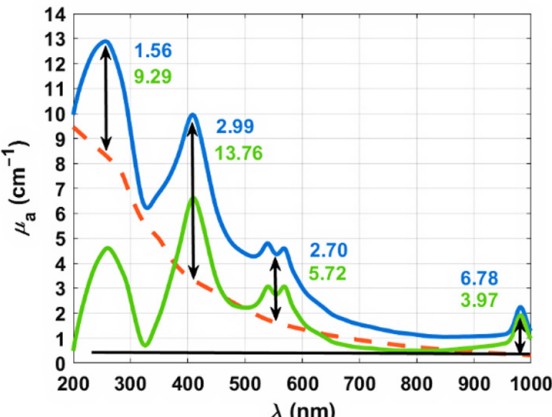

**Figure 9.** Wavelength dependencies for $\mu_a$ of the rabbit pancreas before (blue line) and after (green line) subtracting the absorption of the pigments (orange line).

Figure 9 provides interesting information. First, it indicates that the evaluation of the contents of the main absorbers in the pancreas is deceiving. Such erroneous evaluation is due to the presence of a broadband absorption baseline, which is created by pigments in the pancreas. Only by subtracting such broadband baseline from the $\mu_a(\lambda)$ of the pancreas, a precise evaluation of the contents of the main absorbers can be performed. When reconstructing that absorption baseline from the absorption spectra of melanin and lipofuscin, we observed a similar content for these pigments in the rabbit pancreas. This similar content of both pigments in the pancreas is according to previous reports, where the accumulation of melanin and lipofuscin in similar proportions in tissues has been associated with the aging process [46–48].

Some studies have reported the presence of melanin and lipofuscin in the human and animal pancreas, mainly due to the tissue aging process, but also as a possible indication of the development of diseases [51–55]. Chen et al. reported that melanin was found inside a pancreatic tumor of a 48-year old man with a 10-year follow-up [52]. Wahal et al. reported a rare case of a 61-year-old male patient with a neuroendocrine tumor of the pancreas associated with extensive deposition of melanin [53]. Ref. [54] reports that lipofuscin is also present in human β-cells and that the pancreatic β-cells play a key role in glucose homeostasis by secreting insulin. In his commentary paper [55], Giovanni Di Guardo reports that several studies have found the accumulation of lipofuscin in various neoplasias, including pancreatic tumors. Ref. [56] also reports on the accumulation of lipofuscin in the pancreatic β-cells, and that the injection of artificial lipofuscin in the β-cells of ex vivo pancreatic tissue seems to affect the insulin functionality.

In the face of such literature reports concerning the accumulation of melanin and lipofuscin in the pancreas and their possible association with the development of neoplasia and cancer, it becomes important to perform further studies to allow the discrimination between age-related and disease-related pigment contents. Although the method used in the present study needs excised tissue samples, it is useful to perform such evaluation and allows obtaining discriminated values for the melanin and lipofuscin contents in the pancreas or in other tissues. If an in vivo evaluation of pigment contents is necessary, other approaches must be searched.

## 4. Conclusions

The pancreas is one of the less-studied organs in terms of biophotonics. Using spectral measurements from ex vivo pancreas tissues, we were able to calculate the spectral optical properties of the pancreas in a wide spectral range, between the deep-UV and the NIR. Such spectral optical properties have not been evaluated before and the results presented in this study consist of innovative information about the pancreas needed for research in the field of biophotonics. Further studies that can be performed with this method might

produce differentiated spectral optical properties between healthy and diseased pancreas tissues. Such results would be most valuable for the development of noninvasive optical methods for the diagnosis of pancreas pathologies, which are very lethal.

Although the optical properties obtained in this study show similar $\lambda$-dependencies to the ones previously obtained for other tissues, further analysis of the mean $\mu_a(\lambda)$ of the pancreas allowed us to identify the presence of both melanin and lipofuscin in the pancreas. Since the tissues used in the present study were all retrieved from adult rabbits of similar ages and with no known disease, the identification of similar contents of melanin and lipofuscin in the pancreas seems to indicate that these pigments were accumulated only due to the aging process. A higher accumulation of lipofuscin has been reported in human pathological than in human healthy colorectal mucosa tissues, but such information is unknown for the pancreas. Depending on the availability of both healthy and pathological pancreas tissues, the method used in the present study could be applied to evaluate the melanin and lipofuscin contents and identify if any of these pigments is preferably associated with disease. Both pigments have been reported to be present both in healthy and in diseased pancreas tissues, but their content in each tissue was not accounted for. If such tissues become available in the future, we intend to perform similar studies to quantify the melanin and lipofuscin contents both in the healthy and in the pathological pancreas. Such information will be most useful in the development of new biophotonics methods to diagnose pancreas pathologies that using these pigment contents as disease biomarkers could establish a reliable diagnosis.

**Author Contributions:** Conceptualization, L.M.O. and V.V.T.; methodology, L.M.O.; software, I.S.M. and H.F.S.; validation, L.M.O.; investigation, I.S.M. and H.F.S.; writing–original draft preparation, I.S.M.; writing–review and editing, L.M.O. and V.V.T. All authors have read and agreed to the published version of the manuscript.

**Funding:** The work of I.S.M. was supported by the Portuguese Science Foundation, grant no. FCT-UIDB/151528/2021. The work of H.F.S and L.M.O. was supported by the Portuguese Science Foundation, grant No. FCT-UIDB/04730/2020. The work of V.V.T. was supported by a grant under the Decree of the Government of the Russian Federation No. 220 of 09 April 2010 (Agreement No. 075-15-2021-615 of 4 June 2021).

**Institutional Review Board Statement:** The present study involves measurements from ex vivo animal tissues. The research follows the Declaration of Helsinki and was approved by research review board in biomedical engineering of the Center of Innovation in Engineering and Industrial Technology (CIETI), in Porto, Portugal. Such approval has the number CIETI/Biomed_Research_2021_01.

**Informed Consent Statement:** Not applicable.

**Data Availability Statement:** The data that supports the findings in this study are available from the corresponding author upon reasonable request.

**Conflicts of Interest:** The authors declare no conflict of interest.

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
