# Peer review of "Fast Estimation of the Spectral Optical Properties of Rabbit Pancreas and Pigment Content Analysis"

_photonics, doi:10.3390/photonics9020122_

Round 1

Reviewer 1 Report

You conducted optical analysis for rabbit pancreas, and then tried simple parametric analysis. You researched related studies well, and the analysis was explained thoroughly. But you wrote purpose of this paper to “obtain further information”, we want to know concretely information. Since your result is very clear, I think you can meet my request. Could you reconsider line 96-97? ( or to write focus point on your paper). Abstract and Conclusion are in order, but your discussion was complicated on the last part of results, especially line 339-343, 358-364. Separating sentences are needed.

Line 39 : When is “the mid of the XX century”?

Line 115 : Where is “local”?

Figure 1(c) : Could you described a kind of lamp on the figure?

Figure on 3. Results : What is mean of gradation around full lines?

Author Response

We begin by thanking the reviewer for the suggestions to improve the manuscript. All the corrections indicated below are highlighted in yellow in the corrected manuscript.

  1. Regarding the purpose of the paper, indeed, the reviewer is correct. A precise statement of the purpose of our research was missing. To correct it, we have rewritten the sentence in lines 96-97 as: “With the objective of studying the spectral optical properties of the pancreas and obtain detailed information about its internal composition, tissue ….”
  2. Considering the discussion on the last part of results, especially the text in lines 339-343 and 358-364, where the reviewer recommended the separation of sentences, we have rewritten them as:
    1. Lines 339-343: “Figure 9 provides interesting information. First, it indicates that the evaluation of the contents of the main absorbers in the pancreas is deceiving. Such erroneous evaluation is due to the presence of a broadband absorption-baseline, which is created by pigments in the pancreas. Only by subtracting such broadband baseline from the μa(λ) of the pancreas, a precise evaluation of the contents of the main absorbers can be performed. When reconstructing that absorption baseline from the absorption spectra of melanin and lipofuscin, we observed a similar content for these pigments in the rabbit pancreas. This similar content of both pigments in the pancreas is accordingly to previous reports, where the accumulation of melanin and lipofuscin in similar proportions in tissues has been associated to the tissue aging process [46-48].”
    2. Lines 358-364: “In face of such literature reports concerning the accumulation of melanin and lipofuscin in the pancreas and their possible association with the development of neoplasia and cancer, it becomes important to perform further studies to allow the discrimination between age-related and disease-related pigment contents. Although it needs excised tissue samples, the method used in our study is useful to perform such evaluation and allows obtaining discriminated values for the melanin and lipofuscin contents in the pancreas or in other tissues. If an in vivo evaluation of pigment contents is necessary, other approaches must be searched.”
  3. Considering the comment of the reviewer regarding line 39 – When is the “the mid of the XX century”?, we have corrected that sentence according to the suggestion of Reviewer #3 as: “Since the middle of the XX century the research and…”
  4. Regarding the comment of the reviewer to the word “local” in line 115, we have corrected the sentence as: “Five adult grey rabbits, of similar age near 36 months, were acquired from a breeder near Porto (Portugal) that sells them for consumption.”
  5. To answer the question from the reviewer about the kind of lamp in figure 1(c), we refer that the type of lamp is already described in the third paragraph following Figure 1, as a broad-band deuterium-halogen lamp.
  6. Considering the question of the reviewer regarding “Figure on 3: Results: What is mean of gradation around full lines”, we refer that in all figures in the Results section, the full line is the mean spectrum of measurements or mean of calculations. The gradation surrounding these full lines represents the standard deviation (SD) of those measurements or calculations. Such explanation/reference to the mean/SD spectra is indicated in the text and in figure captions, in section 3.

Reviewer 2 Report

The authors have studied the spectral optical properties of the rabbit pancreas in a broad-spectral range, between 200 nm and 1000 nm. From the analysis of pancreatic tissue of rabbit(s), the authors revealed that the pancreatic tissues not only show the well-known wavelength-dependent optical properties, but further analysis of the spectral absorption coefficient shows that the pancreatic tissues contain pigments, namely melanin and lipofuscin. A detailed spectral analysis of pancreatic tissues makes it possible to implement preventive measures on time, which may help in reducing the risk of pancreatic cancer. The authors have done a thorough study. The manuscript is written well; however, there are a few locations where improvements are necessary. Please find the attached. Annotated texts are my questions, concerns, comments, and suggestions that I recommend the authors consider in their revision before the manuscript goes for publication.

Author Response

We thank the reviewer for his evaluation of our manuscript and the suggestions he has provided in the pdf file to improve it. Special attention was made to those comments and suggestions and the corresponding corrections were made in the manuscript (highlighted in green) as follows:

  1. The sentence in lines 85 and 86 was corrected as suggested.
  2. The text in section 2.1, and the first sentence in section 2.2 were corrected to make it clear that a total of 10 samples, 2 from each rabbit were used in the spectral measurements of the present study.
  3. The text “10x” was replaced by “10 times” throughout the text as recommended.
  4. Some additional text was added in the paragraph above equation (5) to describe the differences between the scattering and the reduced scattering coefficients.
  5. After equation 5, a text was added to explain the meaning of the parameters in that equation.
  6. The wavelength 981 nm was referred some places in the text as a misprint. Following the recommendation of the reviewer, such wavelength was corrected as 980 nm.
  7. We followed the reviewer’s suggestion and replaced the units in the vertical axis of Figure 8.

Reviewer 3 Report

Line 39 should be middle

Line 41 diagnosis

Line 131 is there a pulse rate that can be report for the Xenon lamp?

Line 311 the a needs to be a subscript in your symbol.

After slicing of the samples are they kept frozen or allowed to thaw. So, Frozen or thawed during measurement?

How were the samples mounted on what substrate. Glass slides? I see cuvette for one.

A little more detail could be warranted for Section 2.1

Author Response

The authors thank the reviewer for his comments and suggestions to improve the manuscript. The manuscript has been corrected (corrections highlighted in cyan in the manuscript) according to the suggestions in the following way:

Q1: Line 39 should be middle

R1: The sentence was corrected as: “Since the middle of the XX century…”.

Q2: Line 41 diagnosis

R2: The word has been corrected and the sentence now reads: “…can aid in the establishment of a precise diagnosis, does…”.

Q3: Line 131 is there a pulse rate that can be report for the Xenon lamp?

R3: Indeed, there is a pulse rate for the xenon lamp, but it is controlled by the acquisition software and it depends on the integration time to reach a maximum 150 Hz rate. To clarify, we have written the following text in the description of the setup presented in Figure 1(a). Such description was introduced in the paragraph following Figure 1.

“The Xenon lamp is connected to the spectrophotometer by a Y-cable, which provides power to the lamp and allows to control the flash emission rate through the software up to a limit of 150 Hz.”

Q4: Line 311 the a needs to be a subscript in your symbol.

R4: Thank you. This correction has been made.

Q5: After slicing of the samples are they kept frozen or allowed to thaw. So, Frozen or thawed during measurement?

R5: After slicing, the samples are immersed in saline for ten minutes, before being used in the measurements, to regain the in vivo hydration and warmup to reach the room temperature of 20 °C. A text was introduced at the end of section 2.1 to explain this.

Q6: How were the samples mounted on what substrate. Glass slides? I see cuvette for one.

R6: Indeed, the reviewer is correct. That information was missing in the text. To correct it, we have introduced some text in the description of the setups presented in Figure 1(a) and Figure 1(b) to explain how the tissue samples were fixed to perform those measurements. Please see those descriptions in the first two paragraphs following Figure 1.

Q7: A little more detail could be warranted for Section 2.1

R7: Following the recommendations of all reviewers, text was added to Section 2.1 so that more detail in the collection and preparation of the samples is provided to the reader.